# Using Dried Blood Spots for a Sero-Surveillance Study of Maternally Derived Antibody against Group B Streptococcus

**DOI:** 10.3390/vaccines11020357

**Published:** 2023-02-04

**Authors:** Erick Auma, Tom Hall, Simran Chopra, Sam Bilton, Laxmee Ramkhelawon, Fahimah Amini, Anna Calvert, Gayatri Amirthalingam, Christine E. Jones, Nick Andrews, Paul T. Heath, Kirsty Le Doare

**Affiliations:** 1Department of Biology, Université Claude Bernard Lyon, ENS de Lyon, CNRS, UMR, 69100 Villeurbanne, France; 2Centre for Neonatal and Paediatric Infection, Institute for Infection and Immunity, St George’s, University of London, London SW17 0RE, UK; 3Immunity and Infection, Faculty of Medicine, Imperial College London, London SW7 2BX, UK; 4Neonatal Unit, John Radcliffe Hospital, Oxford University Hospitals NHS Foundation Trust, Oxford OX3 9DU, UK; 5Immunisation and Vaccine Preventable Diseases Division, UK Health Security Agency, London NW9 5EQ, UK; 6Faculty of Medicine and Institute for Life Sciences, University of Southampton, Southampton, SO16 6YD, UK; 7NIHR Southampton Clinical Research Facility and Biomedical Research Centre, University Hospital Southampton NHS Foundation Trust, Southampton SO16 6YD, UK; 8Makerere University—Johns Hopkins University Research Collaboration, Kampala P.O. Box 23491, Uganda; 9Pathogen Immunology Group, UK Health Security Agency, Porton Down, Salisbury SP4 0JG, UK

**Keywords:** Group B Streptococcus, dried blood spot, vaccine, sero-correlate, infant

## Abstract

Vaccination during pregnancy could protect women and their infants from invasive Group B Streptococcus (GBS) disease. To understand if neonatal dried blood spots (DBS) can be used to determine the amount of maternally derived antibody that protects infants against invasive GBS disease, a retrospective case-control study was conducted in England between 1 April 2014 and 30 April 2015. The DBS of cases with invasive GBS disease (*n* = 61) were matched with healthy controls (*n* = 125). The haematocrit, DBS storage temperature, freeze-thaw cycle, and paired serum/DBS studies were set up to optimise the antibody assessment. The samples were analysed using a multiplex immunoassay, and the results were assessed using parametric and nonparametric tests. Antibody concentrations were stable at haematocrits of up to 50% but declined at 75%. DBS storage at room temperature was stable for three months compared with storage from collection at −20 °C and rapidly degraded thereafter. Total IgG levels measured in DBS and paired serum showed a good correlation (*r*^2^ = 0.99). However, due to suboptimal storage conditions, no difference was found in the GBS IgG levels between DBS samples from cases and controls. We have demonstrated a proof of concept that assays utilising DBS for assessing GBS serotype-specific antibodies in infants is viable. This method could be used to facilitate future large sero-correlate studies, but DBS samples must be stored at −20 °C for long term preservation of antibody.

## 1. Introduction

Group B Streptococcus (GBS), also referred to as *Streptococcus agalactiae*, is an opportunistic gram-positive pathogen, and it is the main cause of meningitis and sepsis in neonates and young infants across the globe [1,2]. It is also a significant cause of infections in immunocompromised individuals, pregnant women, and the elderly. Most cases of invasive GBS disease occur between birth and the third month of life [3]. A total of 90,000 fatalities were reported among infants aged below three months, and at least 10,000 surviving infants were reported to suffer from moderate to severe disability following GBS meningitis each year in a global systematic review from 2017 [2]. GBS also accounts for approximately 3.5 million preterm births globally [2]. Vaccinating women aims to protect both women and their infants from infections by enabling the passive transfer of maternal antibodies from the mother to the infant via the placenta and in colostrum and breast milk, thus protecting the infant in the first months of life [4]. Maternal vaccines against GBS are in development, and it is estimated that an effective GBS vaccine could prevent over 300,000 cases and 90,000 infant deaths annually [2]. The relationship between maternally derived antibody and protection against invasive neonatal disease was first demonstrated by Carol Baker in the 1970s; however, the amount of protective antibody required to prevent invasive disease in infants is yet to be elucidated [5].

Defining maternally derived antibody concentrations associated with protection from disease will facilitate the implementation of current and future vaccines. For example, if robust serological correlates of protection for neonatal and infant GBS can be defined, then suitable vaccines might be licensed on this basis, avoiding the need for large and costly efficacy trials; the licensure of meningococcal C and B vaccines were facilitated through such a mechanism [6]. This is particularly true for candidate GBS vaccines, where traditional phase III clinical trials might be logistically and pragmatically difficult to accomplish [7].

To be able to successfully conduct a serological test using a serum sample, one needs to have well-trained phlebotomists for blood collection and adequate laboratory facilities to ensure the transport cold chain and storage of samples. Such temperature requirements limit the ability to obtain samples for disease surveillance in a resource-limited setting. Dried blood spot (DBS) samples are thought to offer a good alternative to serum samples for disease surveillance [8]. The use of DBS as a diagnostic tool dates back to the 20th century and was pioneered by Robert Guthrie for neonatal metabolic disorder screening [9]. The collection of a DBS sample is less expensive, uses readily available and highly portable equipment, and does not require a highly skilled operator to perform, as a finger or heel prick is sufficient to collect the sample. DBS sample collection is also safe and less invasive, and as such, it is more acceptable to participants [10].

Numerous studies have demonstrated the utility and applicability of DBS for testing, such as the retrospective diagnosis of congenital infections [11] and serological detection of a number of infectious diseases [12,13,14,15,16,17,18,19]. Regardless of the extensive use of DBS in a wide range of bioanalysis, its use in detecting maternal IgG in infant sera has yet to be determined. The aim of this project, therefore, was to investigate the use of DBS in determining the concentration of maternally derived antibodies against GBS. In addition, to determine if DBS can be used to measure the minimum level of maternally derived anti GBS IgG antibodies in infant serum associated with the absence of GBS disease in infants < 3 months of age.

## 2. Materials and Methods

### 2.1. Study Design

A retrospective case-control study of infants less than 3 months old was conducted. Infants infected with invasive GBS disease and healthy controls were selected from the National Newborn Screening laboratories from across England [20]. GBS cases were identified by national active surveillance between 1 April 2014 and 30 April 2015 in England. Every GBS case was matched with two controls from the same hospital for sex, gestation, and ethnicity, taken at the same date.

### 2.2. Study Population

There are approximately 700,000 livebirths in the UK [21]. Our active surveillance study for GBS disease estimated the incidence of early-onset disease (EOD), defined as disease occurring within the first 7 days of life, as 0.57 per 1000 livebirths; late-onset disease (LOD) was defined as disease that occurs after day 7 and before day 90 of life, which occurred in 0.37 per 1000 livebirths in the UK [22].

Case definitions: polymerase chain reaction (PCR) positive and/or culture confirmed diagnosis of GBS identified in infants less than 90 days of age in England.

### 2.3. Inclusion and Exclusion Criteria

Consecutive infants with invasive GBS disease over 37 weeks gestation born at any hospital in England. DBS were excluded if the infant had received a blood transfusion according to the information on the DBS card prior to collection of the blood spots, if no consent for research was given on the DBS card, or if no viable blood spots remained at time of collection.

### 2.4. Control Population

Laboratories storing the DBS cards of the infants matching the inclusion criteria randomly selected two DBS from other infants matched for gestation, sex, and ethnicity of the case.

### 2.5. Sample Size

The sample size was set to enable reasonable precision (95% confidence interval width) around an odds ratio of 0.2 when comparing the odds of the GBS antibody concentration being above an antibody threshold in cases and controls. This calculation was based on the assumption that approximately 70% of the healthy uninfected population (controls) have naturally occurring antibodies above a proposed threshold and gave a sample size of 150 cases and 300 controls for a 95% CI around an OR of 0.2 of 0.13–0.30.

### 2.6. Data Collection

New-born screening hospital laboratories in England prepared DBS for the period of January 2014–December 2014. Each study participant was allocated a unique study number. Clinical details were recorded and stored in a secure password-protected database held on a UK Health Security Agency (UKHSA) laptop. Metadata included maternal age, severity of infant illness (defined as the need for Paediatric and Neonatal Intensive Care Unit (PICU/NICU) stay), and clinical outcome following disease (alive well, alive neurological impairment, death). All investigators apart from the UKHSA staff member responsible for collection of the DBS were blinded to the clinical information in the database.

### 2.7. Sample Collection

Once a DBS had been identified by the relevant new-born screening laboratory, the DBS card was checked to ensure at least one viable blood spot per DBS card was left in the laboratory in case future testing was clinically indicated. Two DBS were cut from the card, labelled with the unique study identification number, and placed in a universal container for same-day shipment to the testing laboratory at room temperature. On arrival at St. George’s, University of London (SGUL) laboratory, spots were put into glassine envelopes with desiccant and frozen at −70 °C. Prior to receipt at SGUL, all cards had been stored at room temperature since 2014 (approximately 5 years at room temperature).

### 2.8. Laboratory Methods

#### 2.8.1. DBS Elution

A 3 mm (diameter) punch of each sample was taken and added to Spin-X filter tubes (Costar). Assay buffer (75 µL) (1 × PBS/0.05% Tween 20/0.02% Sodium Azide/0.5% BSA) was added to each tube (1:10 dilution), and the samples were left to elute overnight (20 ± 4 h) at 37 °C. After overnight incubation, the DBS elute sample tubes were spun at 16,000× *g* for 10 min at room temperature. The recovered eluates were transferred to 2 mL cryotubes.

#### 2.8.2. Haematocrit Analysis in DBS

Haematocrit (HCT) levels in a population vary due to diet, age, health, and other factors, and it is important to understand how different levels may influence blood collection as it impacts the spread of blood across the filter paper. Blood group O negative donor red cells were obtained from the local blood donor centre (St. George’s, NHS Foundation Trust). 50 mL of red cells were washed 3 times with an equal volume of saline (0.86 g of NaCl/100 mL of deionised H_2_O). The cells were spun for 5 min at 3000 rpm between each wash. After the final wash, the red cells were spun at 3000 rpm for 10 min to remove any extra saline supernatant. The haematocrit was measured by complete blood cell count and adjusted to >95%. A 1 mL volume of reconstituted blood was generated by mixing 0.5 mL of GBS-vaccinee sera (vaccinated with either monovalent Ia, Ib, II, III, or V, obtained from Carol Baker) with packed red cells at the following concentrations: 25%, 50%, and 75% HCT; 750, 500, and 250 µL of serum were added to 250, 500, and 750 µL of packed red cells, and 50 µL of reconstituted blood was then spotted onto DBS cards (Whatman 903 protein saver cards, Cytiva 10534612, Marlborough, MA, USA) and eluted into assay buffer.

#### 2.8.3. DBS Storage Optimisation

Four DBS cards (five spots per card) were prepared using freshly drawn whole blood from nine volunteers and dried overnight at room temperature (22 °C). The DBS cards were individually placed into separate Ziploc bags with desiccant. One DBS from each donor was stored at −20 °C, 4 °C, 22 °C, or 37 °C. A 3 mm punch was taken from each sample on day 0, 14, and 30, and then every thirty days for the first 6 months and every 6 months up to 1.5 years. The DBS punches were stored in a 1.5 mL tube at −20 °C. At the end of the 1.5-year period, all punches were defrosted into assay buffer and the total IgG concentration was measured by ELISA as follows: we coated 96-well plates (Immulon 4HBX, 3855 Thermo Fisher Scientific, Waltham, MA, USA) with 100 µL of antihuman IgG Fab (I5260 Merck KGaA, Darmstadt, Germany) at 2 µg/mL and left them to incubate overnight at 4 °C. The following day, the plates were washed three times with 1xPBS/0.05% Tween 20 and dried. Blocking buffer (1xPBS/0.05% Tween 20/1% skimmed milk) was added to each well (150 µL), and the plates were incubated for 3 h at room temperature. The plates were then washed again as previously described, and the samples were diluted to 1/800,000 in blocking buffer and added in duplicate to the plates. A serial dilution of human IgG was added as a standard curve, diluted to 0.4 µg/mL in blocking buffer, and serially diluted 3-fold 7 times. The plates were then incubated overnight at 4 °C. The following day, the plates were washed 5 times, and 100 µL of diluted (1/100,000 in 1xPBS/0.05% Tween 20) anti IgG-HRP (A8667, Merck KGaA, Darmstadt, Germany) was added to each well and incubated at room temperature for 3 h. Following a final wash step, 100 µL of TMB was added to each well and incubated for 5 min at room temperature in the dark. 50 µL of sulphuric acid was used to stop the reaction, and the plates were read at 450 nm (BioTek ELx808, Agilent, Santa Clara, CA, USA). 

#### 2.8.4. DBS Stability—Freeze Thaw Cycles

DBS were produced from a pooled positive sample reconstituted with packed red blood cells to a haematocrit of 50%. Once dry, a punch was taken, and the remaining spots were frozen overnight at −20 °C. The following day, the DBS was removed and allowed to defrost at room temperature for 2 h. A punch was taken, and the card was returned to −20 °C. This was repeated four times over four consecutive days. DBS punches were eluted into assay buffer. Three independent replicates were tested at each temperature.

#### 2.8.5. Correlation of Antibodies from DBS and Serum

Serum samples from 18 participants with natural antibodies to GBS capsular polysaccharide (CPS), which were collected from a previous study [23], were reconstituted with packed red blood cells to 50% HCT, spotted onto cards, and dried overnight. A punch from each sample was eluted in assay buffer. Serum and DBS samples were tested side-by-side.

#### 2.8.6. Assessment of GBS Antibodies

Serum and DBS sample eluates from the case control study and DBS method investigations were tested in the multiplex immunoassay using the following methods.

#### 2.8.7. Coupling of Antigens to Bead Sets

A 6-plex pool of beads coupled to CPS, conjugated to Poly-L-Lysine (PLL) for serotypes Ia, Ib, II, III, IV, and V (Pfizer Inc., New York, NY, USA), was prepared as follows: Magplex^TM^ beads (Luminex, Austin, TX, USA) were coupled following the protocol described by Luminex Corporation [24]. CPS-PLL were prepared at a working concentration of 10 µg/mL by dilution into HEPES buffer (Merck KGaA, Darmstadt, Germany), and 500 µL of diluted CPS-PLL were added to the beads. After quality control checks, a stock 6-plex pool was prepared at 5 × 10^5^ in DPBS (Merck KGaA, Darmstadt, Germany).

#### 2.8.8. Multiplex Immunoassay

Samples were tested for IgG levels to 6 GBS serotypes as previously described [25]. Briefly, the samples were incubated overnight with a 6-plex pool of beads, with each bead region coupled to either Ia, Ib, II, III, IV, or V. Samples were diluted to 1/500, 1/5000, and 1/50,000. Each plate included an 11-point standard curve of multivalent vaccinee reference serum (Pfizer Inc., New York, NY, USA) diluted to 1/50 and serially diluted 2.5-fold, and two wells containing assay buffer acted as blank controls. The next day, the plates were washed with 200 µL of PBS/Tween (1xPBS/0.05%Tween/0.02%/Sodium Azide, pH 7.2) using a plate washer (Tecan Hydrospeed, Tecan, Reading, UK) with a magnetic base to retain the beads. Secondary antibody was added (R-Phycoerythrin Goat Anti-Human IgG Fcy specific, Jackson Laboratories 109-115-098, Jackson ImmunoResearch, Ely, UK), at 1/500 (50 µL) to each well, and the plates were incubated for 90 (±15) minutes at room temperature under constant shaking. Following the incubation, the plates were washed again, and 100 µL of wash buffer added to each well. The plates were read on a Bio-plex 200 (Bio-Rad Laboratories, Hercules, CA, USA) at high RP1 (high photomultiplier tube voltage). The results were reported as the median fluorescent intensity (MFI), analysing a minimum of 50 beads per bead set.

### 2.9. Data Processing and Analysis

A statistical analysis was performed using GraphPad Prism version 8.0.2 (GraphPad Software Inc., San Diego, CA, USA). The means and 95% confidence intervals or median with interquartile range were calculated for each serotype depending on whether the data was normally distributed. Parametric (*t*-test) and non-parametric tests (Friedman, Mann–Whitney) were used to test for differences between antibody concentrations. A Deming regression was performed to estimate the systematic difference between antibody concentrations measured in the serum and DBS. A conditional logistic regression was used to assess the relationship between serotype-specific antibody titre and case/control status using cut-offs at 0.3, 1, 3, and 10 ug/mL to represent a range on the log scale. The analysis was undertaken in three steps: just serotype III cases, all cases, and all cases except those known to be other serotypes. The relationship between serotype-specific antibody titre in cases and controls was also examined using reverse cumulative distribution (RCD) curves.

## 3. Results

### 3.1. Case-Control Study

We identified 211 DBS from 805 cases of GBS in England between January and December 2014 from the British paediatric surveillance unit (BPSU) study [22]. DBS were retrieved from a total of 483 participants comprising 161 cases of invasive GBS disease and 322 controls. Of the 483 individual samples, 186 DBS samples (38.51%) (61 cases and 125 controls) from three national screening laboratories were eluted and analysed, the remainder did not elute (Appendix A). Gestational age and gender data was available for 98% of participants (61 cases, 125 controls). The baseline characteristics of the study participants are summarised below in Table 1. Prior to conducting further analyses on these samples, we sought to investigate potential parameters that would influence our results, this included haematocrit, storage temperature, freeze–thaw cycles, and the correlation between dried blood spots and serum from healthy adult donors.

### 3.2. High Haematocrit May Affect Antibody Concentrations Readouts

We observed a significant difference between laboratory reconstituted DBS samples with 75% haematocrit compared with 25% and 50% haematocrit for serotypes Ia, Ib, II, III, and V (*p* < 0.01). (Figure 1A).

### 3.3. Storage Temperatures Affect Antibody Stability

There were marked declines in total IgG antibody concentrations at all temperatures above 4 °C within the first three months (*p* < 0.01) (Figure 1B). Compared with the baseline, the concentration of IgG for DBS stored at room temperature declined by 17.1% (*p* = 0.01) after 5 months, 20.2% (*p* < 0.01) after 6 months, and 33.8% (*p* < 0.001) after 12 months. The DBS samples stored at 4 °C had stable IgG concentrations until the fifth month, after which declines of 20.0% in IgG was observed in comparison to the baseline (*p* < 0.01). The DBS samples stored at −20 °C had stable IgG concentrations until the sixth month and had declined by 9.3% at 12 months (*p* = 0.23) in comparison to day 0. The DBS were stable for up to three freeze–thaw cycles, with a non-significant decline in IgG levels observed after the fourth cycle (Figure 1C).

### 3.4. Correlation between Freshly Prepared DBS and Paired Serum Samples

A strong correlation was found between laboratory-prepared DBS and serum concentrations for a subset of samples (*n* = 18) (R^2^ = 0.99) (Figure 1D). The 95% confidence interval (CI) for the intercept included 0, and the 95% CI for the slope included 1, indicating no systematic differences between the two sample collection techniques.

### 3.5. Antibody Concentrations in Invasive GBS Cases and Controls

We observed no difference in antibody concentration between invasive GBS cases and controls from the 186 eluted DBS samples. On collection, there was a limited number of GBS cases for which a specific serotype was documented, which hampered serotype-specific analysis. Table 2 describes the geometric mean antibody concentrations for GBS cases with a documented GBS serotype. We analysed the antibody concentrations for LOD serotype III GBS disease, the only serotype for which there were sufficient cases compared with healthy controls, but this was not significant (*n* = 6 cases; *p* = 0.46) (Figure 2). The mean age at onset of serotype III LOD was 14.44 (95% CI 12.37–16.52). We were unable to ascertain the interval between card collection time and onset of LOD.

### 3.6. Logistic Regression to Assess Cut-Offs and Reverse Cumulative Distribution of Antibodies against Serotype III for Cases and Controls

Antibody distributions are similar in cases and controls across the range of cut-offs considered (Table 3 and Figure 3). For the cut-off at one, the odds ratios are below one (indicative of a protective effect of higher antibody levels); however, the 95% CIs overlap one, and the differences are not significant. The numbers are very small (so power is too low) when only considering serotype III cases. Adding other cases, many of which are likely to be serotype III, improves power, but this analysis is contaminated by non-serotype-III cases.

## 4. Discussion

This study demonstrates that DBS may be a useful tool to determine minimum maternally derived antibodies required to protect against GBS disease. However, although we found that DBS made from reconstituted sera and optimally stored had good correlation with sera, the degradation of the clinical study samples meant that we were unable to demonstrate a correlate of protection. Nevertheless, we were able to measure GBS specific IgG in the few samples that successfully eluted, and we did see a differential effect of serotype III antibodies in the cases compared with the controls in our RCD plots, although this was not significant. Other studies with higher numbers of cases have found that antibody concentrations above a threshold may be protective [25,26].

Several factors may have affected antibody yield and stability in the clinical study, including unknown haematocrits and card storage conditions [27]. A concern about DBS is the impact of haematocrit on the spreading of blood across a filter paper, as the haematocrit affects blood viscosity and therefore spot size and the volume of blood in a punched disc may vary. The haematocrit in neonates fluctuates with gestational and postnatal age; with the advancing of gestation, the mean haematocrit level progressively increases, and in the first 28 days after birth, a more or less linear decrease in haematocrit occurs [28]. This is caused by rapid body growth resulting in haemodilution, a shorter lifespan of the red blood cells, poor iron stores, and the liver being the primary source of erythropoietin production in the first weeks of life [29]. In addition, the haematocrit in neonates is known to have wider variability (28–67%) [30,31] compared with the haematocrit in adult subjects (38–46%) [32]. We have shown that antibodies eluted from DBS are stable up to a haematocrit value of 50%; however, at 75%, the amount of antibody eluted reduces, suggesting that a correction factor may be needed for neonatal blood spots.

Storage conditions are particularly vital for a retrospective study such as ours, which was conducted on DBS samples stored for over two years after collection at room temperature. We were only able to recover 40% of DBS samples due to significant degradation at room temperature at sample collection facilities. A review by Amini et al. identified a key concern regarding adequate storage conditions, finding that DBS samples stored at room temperature were stable for up to 28 days, but that the optimum temperature for long-term stability of DBS samples was likely to be −20 °C with silica desiccant [27]. This supports the temperature stability work conducted here, which demonstrated that IgG levels remained stable at −20 °C over 12 months. DBS samples are stored at room temperature in many countries, which may hamper the utility of this sample collection method in serological studies.

Serum is generally considered the “gold standard” sample to measure protective immunity [33]. DBS samples offer an attractive alternative to serum antibody testing, and our experiments with DBS made from anti GBS IgG positive serum reconstituted with red blood cells have shown that there is no significant differences between serum and DBS, as has been seen in other studies [34,35]. However, the strength of the correlation may be affected by the antigen target, thus requiring careful investigation for each new target of interest [27].

One of the study strengths was that the cases and controls were matched for potential confounding variables known to bias or influence antibody concentrations. We recognise that our study had several limitations; hence, the interpretation of the results should be made with caution. Due to the time lapse between the original study and card collection, not all cards could be located; hence, the number of samples available for analysis was reduced. We also lacked serotype information on many cases, which hampered our results. Finally, our study measured natural antibodies, and these may be at a lower level compared with vaccinated sera.

## 5. Conclusions

We were unable to demonstrate a difference in antibody concentrations between cases and controls mainly due to poor sample storage conditions resulting in few viable cards. For DBS to be a useful resource for sero-correlate studies, it is vital that they are correctly stored immediately after collection. The authors suggest storage at −20 °C as quickly as possible after the spots have dried. The lack of clear guidance in England on storage conditions impacts our ability to use such a valuable resource to potentially answer a range of important questions.

## Figures and Tables

**Figure 1 vaccines-11-00357-f001:**
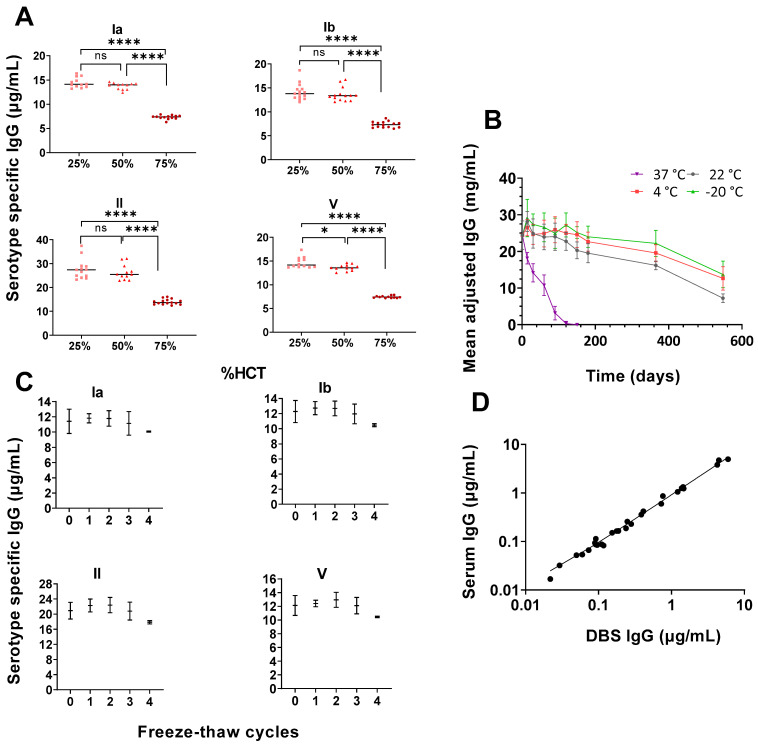
(**A**–**D**). (**A**): Serotype-specific (Ia, Ib, II and V) IgG concentrations (µg/mL) with haematocrit levels of 25%, 50%, and 75%. * *p* = 0.010, **** *p* ≤ 0.0001. (**B**): The stability of the mean-adjusted total IgG levels in DBS stored at different temperatures over 18 months (mean ± S.D). (**C**): Median and interquartile range of serotype specific IgG levels (ug/mL) (Ia, Ib, II, and V) in eluates taken from laboratory-prepared DBS after 0–4 freeze thaw cycles. (**D**): Correlation between paired laboratory spiked DBS and serum samples (*n* = 18). For simplicity, serotypes III and IV are not shown; however, the serotypes included in the figure are illustrative of the general trend.

**Figure 2 vaccines-11-00357-f002:**
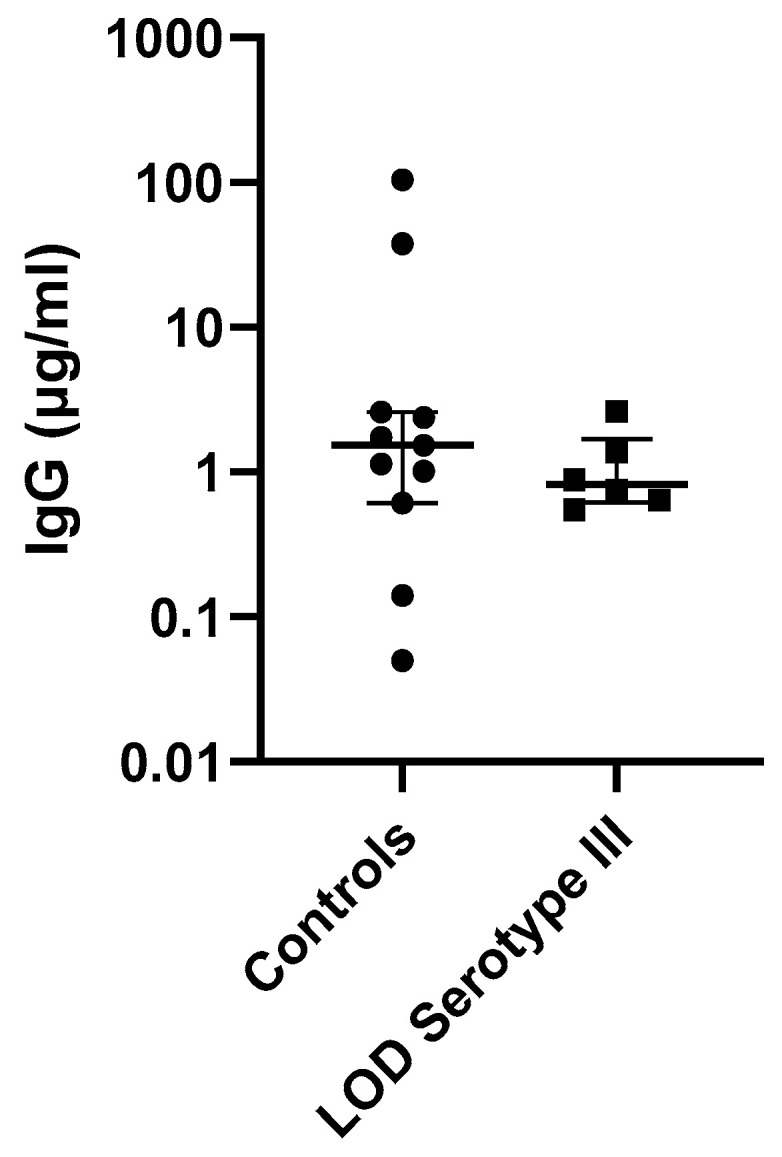
IgG antibody concentrations for serotype III LOD cases vs. controls. Median and interquartile ranges of IgG levels shown. On the left, controls (*n* = 11); on the right, cases (*n* = 6).

**Figure 3 vaccines-11-00357-f003:**
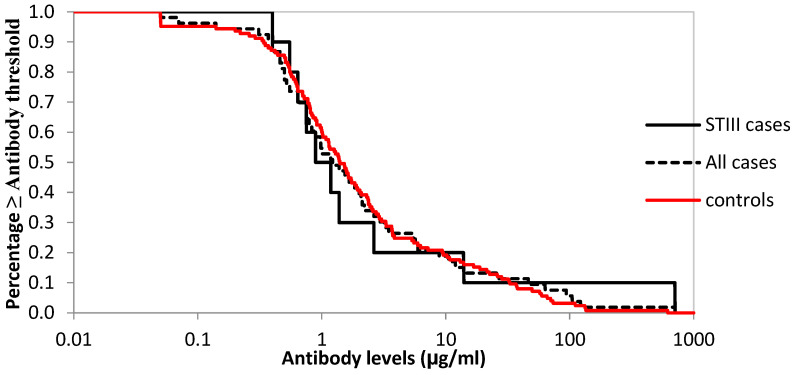
Reverse cumulative distribution curve of serotype III antibodies. The distribution of concentration of maternally transferred GBS specific serotype III antibodies from DBS blood samples of infants born in England. Note that the RCD curve for all cases except those serotyped as not serotype III is very similar to the serotype III cases curve.

**Table 1 vaccines-11-00357-t001:** Characteristics of cases and controls.

Characteristics	Case, *n* (%)	Controls, *n* (%)	Total, *n* (%)
DBS analysed	61 (32.79%)	125 (67.20%)	186 (100)
Male	33 (32.04%)	70 (67.96%)	103 (55.38)
Female	26 (33.77%)	51 (66.23%)	77 (41.40)
Missing	2 (16.67%)	4 (83.33%)	6 (3.23%)
Mean gestation age (weeks)	38.58 (37–41)	38.90 (38–41)	

**Table 2 vaccines-11-00357-t002:** Geometric mean antibody concentrations in the cases and controls (µg/mL).

Serotype	Ia	Ib	II	III	IV	V
Cases, homotypic serotype	183.01 *	NA **	NA **	2.29 *** (0.46–11.47)	NA **	NA **
Case, any serotype	1.71 (0.75–3.92)	1.03 (0.55–1.94)	5.42 (2.36–12.45)	1.92 (1.19–3.12)	0.02 (0.02–0.02)	1.37 (0.70–2.69)
Controls	2.89 (1.58–5.30)	1.00 (0.64–1.59)	5.55 (3.26–9.47)	2.00 (1.45–2.80)	0.02 (0.02–0.02)	1.05 (0.67–1.66)

* 2 cases of Ia, ** 1 case of Ib, II, IV, and V; *** 10 cases of serotype III. NA: no homotypic serotypes.

**Table 3 vaccines-11-00357-t003:** Conditional logistic regression results using serotype (ST) III titres with cut-offs at 0.3, 1, 3, and 10 and different case definitions.

Cut-Off	Case Inclusion	Cases		Controls		Odds ≥ Cut in Cases vs. Controls	
		<Cut-Off	≥Cut-Off	<Cut-Off	≥Cut-Off	Conditional OR (95% CI)	*p*-Value
0.3	STIII	0	10	11	114	n/a	
	STIII and ST not known	3	50	11	114	1.61 (0.39–6.59)	0.51
	All	3	56	11	114	1.81 (0.46–7.17)	0.40
1	STIII	5	5	49	76	0.78 (0.15–4.20)	0.77
	STIII and ST not known	23	30	49	76	0.66 (0.29–1.48)	0.31
	All	27	32	49	76	0.67 (0.32–1.41)	0.29
3	STIII	8	2	86	39	0.67 (0.07–6.41)	0.73
	STIII and ST not known	37	16	86	39	0.93 (0.45–1.95)	0.85
	All	42	17	86	39	0.86 (0.43–1.74)	0.68
10	STIII	8	2	101	24	1.00 (0.09–11.03)	1.00
	STIII and ST not known	44	9	101	24	0.78 (0.32–1.94)	0.60
	All	49	10	101	24	0.77 (0.33–1.77)	0.54

## Data Availability

The data presented in this study are openly available in FigShare at 10.6084/m9.figshare.21976187.v1.

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
