# Peer review of "Using Dried Blood Spots for a Sero-Surveillance Study of Maternally Derived Antibody against Group B Streptococcus"

_vaccines, 2023, doi:10.3390/vaccines11020357_

Round 1
Reviewer 1 Report
The authors were unable to demonstrate a difference in antibody concentrations between cases and controls mainly due to poor sample storage conditions resulting in few viable cards.
For DBS to be a useful resource for serocorrelate studies, it is vital that they are correctly stored immediately after collection.
The lack of clear guidance in England on storage conditions impacts on our ability to use such a valuable resource to potentially answer a range of important questions.

Author Response
Thank you for your comments.
Reviewer 2 Report
This manuscript describes potentially important idea and findings, showing possibility to measure maternally derived antibody to GBS using DBS. It is interesting to find that antibody concentrations in serum and DBS were correlated, although specification and quantification of antibody could not be performed. Because this study is based on authors' unique idea to use DBS and most readers may not be familiar with methods and way of data evaluation, it is somewhat difficult for readers to understand the flow of the story that were written. Therefore, this reviewer suggests to check and describe more details and/or rephrase explanations in Results section, so that readers to understand better the story of this research, i.e., to show what was attempted for which purpose, and what was successful or not.
Minor comments are as follows.
1. Title : Abbreviation "(GBS)" is not necessary to add.
2. Abstract: Is the iGBS commonly used in other research articles? It is confusing to use iGBS as invasive GBS disease, because GBS is abbreviation of "Group B Streptococcus". Consider other word like "iGBSD" or other used in published literatures.
3. Abstract. line 26, "DBS and paired serum showed good correlation". This sentence is too simple to understand what point was correlated. Add some words to be understandable.
4. Abstract, line 26. Add "However, " before "Due to ....", if it fits the context. The sentence "Due to sub optimally stored DBS from..." should be changed to "due to suboptimal storage of DBS from iGBS cases and control, ..." and so on.
5. line 35: "Streptococcus agalactiae" should be italicized.
6. line 249, 3.4, section title: "Good" should be deleted. Section title should be just item of analysis. Good or not should be described in text.
7. line 625: Is "LOD" an abbreviation of "late onset"? LOD appears often thereafter, but it is difficult to understand. Define LOD exactly.
8. Table 2. It is better to add "serotype" in the rightmost column, for uppermost line. Table 3: If STIII means serotype III, it should be written in footnote.
9. line 37: [1], [2] should read [1,2]. Similarly, line 310 should read "[30,31]", line 299 should read [25,26]. Check whole text carefully.
Author Response
Response to reviwer 2 comments
Comment 1: Title: Abbreviation "(GBS)" is not necessary to add.
Response: We have made an amendment to the manuscripts title, line 3.
“against Group B Streptococcus.”
Comment 2: Abstract: Is the iGBS commonly used in other research articles? It is confusing to use iGBS as invasive GBS disease, because GBS is abbreviation of "Group B Streptococcus". Consider other word like "iGBSD" or other used in published literatures.
Response: We have changed this to invasive GBS disease throughout.
“Vaccination during pregnancy could protect women and their infants from invasive Group B Streptococcus (GBS) disease.”
Comment 3: Abstract. line 26, "DBS and paired serum showed good correlation". This sentence is too simple to understand what point was correlated. Add some words to be understandable.
Response: Abstract, line 27 has been amended to add clarity.
“Total IgG levels measured in DBS and paired serum showed good correlation (r2 = 0.99).”
Comment 4: Abstract, line 26. Add "However, " before "Due to ....", if it fits the context. The sentence "Due to sub optimally stored DBS from..." should be changed to "due to suboptimal storage of DBS from iGBS cases and control, ..." and so on.
Response: Abstract, line 28 has been amended to add clarity.
“However, due to sub optimal storage conditions, no difference was found in GBS IgG levels between DBS samples from cases and controls.”
Comment 5: line 35: "Streptococcus agalactiae" should be italicized.
Response: We have made an amendment line 38 in the manuscript.
“Streptococcus agalactiae”
Comment 6: line 249, 3.4, section title: "Good" should be deleted. Section title should be just item of analysis. Good or not should be described in text.
Response: We have made an amendment 3.4, section title in the manuscript.
“Correlation between freshly prepared DBS and paired serum samples”
Comment 7: line 265: Is "LOD" an abbreviation of "late onset"? LOD appears often thereafter, but it is difficult to understand. Define LOD exactly.
Response: We have defined late onset disease as disease occurring after day 7 and before day 90 of life. Line 93-96 have been updated to give definitions of early and late onset disease. From then on late onset disease is referred to as LOD.
“Our active surveillance study for GBS disease estimated the incidence of early-onset disease (EOD), defined as disease occurring within the first 7 days of life, as 0·57 per 1000 livebirths and for late-onset disease(LOD), defined as disease that occurs after day 7 and before day 90 of life, as 0·37 per 1000 livebirths in UK [22].”
Comment 8: Table 2. It is better to add "serotype" in the rightmost column, for uppermost line. Table 3: If STIII means serotype III, it should be written in footnote.
Response: We have updated Table 2 and Table 3 in the manuscript.
Comment 9: line 37: [1], [2] should read [1,2]. Similarly, line 310 should read "[30,31]", line 299 should read [25,26]. Check whole text carefully.
Response: Updated and all references checked.
Line 37 “neonates and young infants across the globe [1,2].”
Line 376 “that antibody concentrations above a threshold may be protective [25,26].”
Line 387 “neonates is known to have wider variability (28–67%) [30,31] compared to the haematocrit”
Reviewer 3 Report
The manuscript aimed to study whether dried blood spots could be used to determine the amount of maternally derived antibody that can protect the infants against invasive Group B Streptococcus. Their main findings are that the storage conditions of the dried blood spots are important for later analysis. The researchers didn't manage to use this method to study 5 years old DSB samples stored at room temperature. This might either be due to improper storage or not optimal elution. Could recent DSB samples of GSB infant cases be used in this technique? Also, it would be an advantage if the researchers can prove that their method can be used to detect serotypes that have been transferred from the mother to the infant.
Abstract:
Line 17: First time used, spell the full name of GBS.
Line 18: I would suggest writing "can be used" instead of "were used"
In the last sentence of the abstract, please state which storage condition is optimal.
Introduction: Please use italics for bacterial names in Latin.
Material and Methods:
A verb is lacking in the sentence of line 82-83: I would suggest writing "was selected"
State where is the "National Newborn Screening laboratories".
Line 84: small letter in "cases"
In abstract it states between "2014 and 2015" while in the methods , they state during the year 2014. Please correct to right statement.
Section 2.5: Please state what was defined as the antibody threshold.
Section 2.7: If the samples have been stored at room temperature for 5 years, would a sudden storage at minus 70 actually destroy the antibody due to sensitivity to changes in the temperature? Are the places where the DBS is stored from the day at collection to testing at minus 20 or minus 70?
Line 132: Please use mu (μ) instead of u.
Line 133: What do you mean with a 1/75 dilution? You added 75 microliters of the assay buffer in order to extract the antibodies from the DBS. Maybe you intend that the DBS was one microliter and thus diluted 1:75 upon addition of the assay buffer? Why was the elution done at 37°C and not at 4°C? Why was the centrifugation done at room temperature and not at 4°C?
Did you try other elution methods? (Confer doi:10.1111/tmi.12259).
Could the bacteria in the blood sample affect the elution of IgG?
Section 2.8.2: Please add some words about the aim of this method.
Line 152: Instead of "everyone" I would suggest writing "each donor"
The grade sign should be correct in the text to °.
Line 162: Why was such a high dilution factor of 800,000 used?
The assay detects IgG in general and not specifically antibodies to GBS.
Line 172: Please write "made" instead of "created"
Line 179: Define CPS first time mentioned.
Line 180: Why was it necessary to add packed red cells for the assay?
Line 185 should end in a colon ":"
The same for line 188: after "as follows" add colon instead of semicolon.
A space should be added before the bracket of the reference. (Several places in the text).
Line 192: the "5" should be in superscript.
Line 200: Correct to "At the following day"
Line 202: Correct to "under constant shaking"
Line 201: The exact name of the secondary antibody should be stated (not enough the catalog number). What was the dilution factor of the secondary antibody?
Section 2.8.8: Please describe how the beads were washed in the plates.
Did you have access to fresh DBS from GBS infant cases?
Line 226: correct to: "were eluted and analyzed"
I think the lines 229-230 should be in the first result section and then the data from the sample study.
In Table 1: The number of "missing" is not compatible with the number of cases. It should be 2 and 4 and not 1 and 5.
According to section 3.3 – there were no changes of detecting antibodies in 5 year old samples. How would the researchers suggest conserving the IgG of the DBS samples?
The labels in Figure 1 should be larger in order to be readable. In C: What about the two other serotypes? (There were six and only 4 are presented).
For Figure 1A: If you didn't add red blood cells, what would the data be?
Table 2: Define NA.
Could your method be proved to be efficient in studying GBS serotypes in recent DBS samples?
One of the aims of the research was to show that the method could be used to determine the presence of GBS serotypes in infants of vaccinated versus non-vaccinated pregnant mothers using DBS. However, data are lacking concerning this issue. This would have been a great advantage.
The abbreviation STIII should be defined in the text.
Figure 3 use the Greek mu letter instead of u.
How were the y-axis values calculated?
Author Response
Response to reviewer 3 comments
Comment 1: Line 17: First time used, spell the full name of GBS.
Response: Line 18 has been updated.
“invasive Group B Streptococcus (GBS) disease.”
Comment 2: Line 18: I would suggest writing "can be used" instead of "were used"
Response:. Abstract has been updated to add clarity. Lines 18-20 have been amended
“To understand if neonatal dried blood spots (DBS) can be used to determine the amount of maternally derived antibody that protects infants against invasive GBS disease a retrospective case-control study was conducted in England between the 1st April 2014 and the 30th of April 2015.”.
Comment 3: In the last sentence of the abstract, please state which storage condition is optimal.
Response: Abstract has been updated. Line 32-33.
“This method could be used to facilitate future large sero-correlate studies, but DBS samples should be stored at -20°C for long term preservation of antibody.”
Comment 4: Introduction: Please use italics for bacterial names in Latin.
Response: We have made an amendment line 38 in the introduction.
“Streptococcus agalactiae”
Material and Methods:
Comment 5: A verb is lacking in the sentence of line 82-83: I would suggest writing "was selected"
Response: Line 85-87 has been updated.
“A retrospective case-control study of infants less than 3 months old was conducted. Infants infected with invasive GBS disease and healthy controls were selected from the National Newborn Screening laboratories from across the England[20].”
Comment 6: State where is the "National Newborn Screening laboratories".
Response: We have clarified and added the reference to the website http://newbornscreening.org/site/laboratory-directory.asp, line 87.
“the National Newborn Screening laboratories from across the England[20].”
Comment 7: Line 84: small letter in "cases"
Response: This has been amended, line 87.
Comment 8: In abstract it states between "2014 and 2015" while in the methods, they state during the year 2014. Please correct to right statement.
Response: This has been amended. Line 88-89
“A retrospective case-control study was conducted in England between the 1st April 2014 and the 30th of April 2015”
Comment 9: Section 2.5: Please state what was defined as the antibody threshold
Response: A specific threshold was not assumed, as this is what we are investigating. Multiple threshold were examined and we were looking at a scenario where a threshold exists where you get an odds ratio of about 0.2.
Comment 10: Section 2.7: If the samples have been stored at room temperature for 5 years, would a sudden storage at minus 70 actually destroy the antibody due to sensitivity to changes in the temperature? Are the places where the DBS is stored from the day at collection to testing at minus 20 or minus 70?
Response: It is common practice to store samples for antibody testing at -700C, from room temperature. Antibodies are resistant to these storage conditions, the fresh DBS used in this study were all frozen at -700C from room temp, and both total and specific IgG could be measured. No national screening laboratory in the UK stores DBS from day of collection at either -20 or -70 C. Laboratories in Scotland store samples at -20C from collection
Comment 11: Line 132: Please use mu (μ) instead of u.
Response: Updated.
Line 137 “75µl of assay buffer”
Comment 12: Line 133: What do you mean with a 1/75 dilution? You added 75 microliters of the assay buffer in order to extract the antibodies from the DBS. Maybe you intend that the DBS was one microliter and thus diluted 1:75 upon addition of the assay buffer? Why was the elution done at 37°C and not at 4°C? Why was the centrifugation done at room temperature and not at 4°C?
Response: The dilution factor has been updated; the DBS elution is in fact a dilution of 1:10: 7.5µl of blood in the DBS eluted into 75µl of buffer. Line 138.
“75µl of assay buffer (1xPBS/0.05% Tween 20/0.02% Sodium Azide/0.5 % BSA) was added to each tube (1:10 dilution) and the samples left to elute overnight (20± 4 hours) at 37°C.”
Comment 13: Why was the elution done at 37°C and not at 4°C? Why was the centrifugation done at room temperature and not at 4°C?
Response: The elution conditions were used based on prior elution method comparisons.
Comment 14: Did you try other elution methods? (Confer doi:10.1111/tmi.12259).
Response: The elution method was chosen based on previous testing and those most commonly used in the literature. We were not aware of the application of ammonium to improve blood elution from poorly stored DBS, however this paper does not demonstrate if IgG can be measurable from the fixed DBS eluted using ammonium. This will be something our lab will investigate in the future
Comment 15: Could the bacteria in the blood sample affect the elution of IgG?
Response: This is a possibility, typically our lab finds samples from non-sterile sites (breast milk and vaginal secretions) are susceptible to IgG degradation due to the presence of bacteria. However, we feel that despite other factors, the poor storage contributed significantly to our inability to measure IgG accurately and would certainly exacerbate bacterial degradation.
Comment 16: Section 2.8.2: Please add some words about the aim of this method.
Response: We have updated the Section 2.8.2. Line 143-145.
“Haematocrit (HCT) levels in a population vary due to diet, age, health and other factors, it is important to understand how different levels may impact blood collection as it impacts the spread of blood across the filter paper.”
Comment 17: Line 152: Instead of "everyone" I would suggest writing "each donor"
Response: Updated.
Line 160-161: “One DBS from each donor was stored at -200C, 40C, 220C, or 370C”
Comment 18: The grade sign should be correct in the text to °.
Response: Updated
Line 163 “DBS punches were stored in a 1.5ml tube at -200C“
Comment 19: Line 162: Why was such a high dilution factor of 800,000 used?
Response: This experiment measured total IgG concentrations, which is typically found in very high concentrations when compared to IgG specific to an antigen. A high dilution is required to bring the concentration of IgG to within the range of the assay.
Comment 20: The assay detects IgG in general and not specifically antibodies to GBS.
Response: Yes, this experiment measured total IgG as GBS specific antibodies are often very low in natural samples. Therefore, it was deemed better to measure total IgG in donor serum. Although not ideal, it is still possible to generalise about the stability of GBS specific antibodies.
Comment 21: Line 172: Please write "made" instead of "created"
Response: Updated
Line 180 “DBS were made from a pooled positive sample reconstituted with packed red blood’
Comment 22: Line 179: Define CPS first time mentioned.
Response: Updated
Line 187: “Serum samples from 18 participants with natural antibodies to GBS capsular polysaccharide (CPS) collected from”
Comment 23: Line 180: Why was it necessary to add packed red cells for the assay?
Response: Archived serum samples were used in this experiment. To generate a dried blood spot the serum is first reconstituted as blood, using packed red blood cells to generate a ~50% HCT. This is the spotted onto a card to generate the blood spot. This simulates spotting of blood directly from venous blood.
Comment 24: Line 185 should end in a colon ":"
Response: Line 185 is the end of a sentence, no change required.
“A punch was eluted in assay buffer and serum and DBS samples were tested side by side.”
Comment 25: The same for line 188: after "as follows" add colon instead of semicolon.
Response: Updated
Line 196 “Ia, Ib, II, III, IV and V was prepared as follows: MagplexTM beads (Luminex, Austin”
Comment 26: A space should be added before the bracket of the reference. (Several places in the text).
Response: Updated
Comment 27: Line 192: the "5" should be in superscript.
Response: Updated, line 200.
“checks, a stock 6-plex pool was prepared at 5x105 in DPBS”
Comment 28: Line 200: Correct to "At the following day"
Response: Changed to “The next day”
Line 208 “two wells containing assay buffer acted as blank controls. The next day the plates were”
Comment 29: Line 202: Correct to "under constant shaking"
Response: Updated
Line 213 “incubated for 90 (±15) minutes at room temperature, under constant shaking.”
Comment 30: Line 201: The exact name of the secondary antibody should be stated (not enough the catalog number). What was the dilution factor of the secondary antibody?
Response: Updated
“50µl of secondary antibody (R-Phycoerythrin Goat Anti-Human IgG Fcy specific, Jackson Laboratories 109-115-098), at 1/500,was added to each well”
Comment 31: Section 2.8.8: Please describe how the beads were washed in the plates.
Response: Updated the section 2.8.8. Line 208-211.
“The next day the plates were washed with 200µl of PBS/Tween (1xPBS/0.05%Tween/0.02%/Sodium Azide, pH 7.2) using a plate washer (Tecan Hydrospeed, Tecan UK) with a magnetic base to retain the beads.”
Comment 32: Did you have access to fresh DBS from GBS infant cases?
Response: Not at the time of the study.
Comment 33: Line 226: correct to: "were eluted and analyzed"
Response: Updated
Line 237 “125 controls) from three national screening laboratories were eluted and analysed, the”
Comment 34: I think the lines 229-230 should be in the first result section and then the data from the sample study.
Response: We included section 3.1 as it describes the overall purpose of the study. However, as the elution of the DBS failed we describe the investigations into the use of DBS and important parameters.
Comment 35: In Table 1: The number of "missing" is not compatible with the number of cases. It should be 2 and 4 and not 1 and 5.
Response: Table 1 updated.
Comment 36: According to section 3.3 – there were no changes of detecting antibodies in 5 year old samples. How would the researchers suggest conserving the IgG of the DBS samples?
Response: A line has been added to the conclusion that includes a storage suggestion. Line 358
“the authors suggest storage at -20°C as quickly as possible after spots have dried.”
Comment 37; The labels in Figure 1 should be larger in order to be readable. In C: What about the two other serotypes? (There were six and only 4 are presented).
Response:. The Figure 1 labels has been updated. For simplicity, only 4 key serotypes have been included in the Figure 1. The ones chosen to be included are illustrative of the overall trends.
Comment 38: For Figure 1A: If you didn't add red blood cells, what would the data be?
Response: The antibody levels would be the same as the 25 and 50% HCT.
Comment 39: Table 2: Define NA.
Response: We have updated Table 2 with NA definition.
Line 349 “Line *2 cases of Ia, **1 case of Ib, II, IV and V; ***10 cases of serotype III. NA: no homotypic serotypes.”
Comment 40: Could your method be proved to be efficient in studying GBS serotypes in recent DBS samples?
Comment 41: We demonstrate that DBS collected from volunteers and stored at -20C have measurable antibody for at least a year. We would suggest that these finding can be extrapolated to GBS specific antibody. Additionally, although most of the DBS were of too poor quality to measure antibody, we have demonstrated that in the few samples that eluted successfully, IgG specific to GBS antigens is measurable (table 2), I have added extra detail to make this point in the discussion (line 312). Unfortunately, the sample size was too low to detect antibody differences after poorly eluted samples were removed from the analysis. Finally, we also demonstrate that serum samples positive for GBS specific IgG can be reconstituted as blood and used to generate a DBS can be eluted successfully and the GBS specific IgG levels correlate well with those seen in the original sera.
Comment 42: One of the aims of the research was to show that the method could be used to determine the presence of GBS serotypes in infants of vaccinated versus non-vaccinated pregnant mothers using DBS. However, data are lacking concerning this issue. This would have been a great advantage.
Agreed, unfortunately the samples were of poor quality and antibody could not be eluted. Instead, we investigated the utility of DBS more generally, exploring various factors that could limit their use.
Comment 43: The abbreviation STIII should be defined in the text.
Response: Updated.
Comment 44: Figure 3 use the Greek mu letter instead of u.
Response: Updated.
Comment 45: How were the y-axis values calculated?
Response: At each antibody level from 0.01 to 1000 ug/l, in steps of 0.01 log10 ug/l, the proportion of samples with a titre greater than or equal to that level is calculated. This is then plotted using a line plot with the x-axis as the titre and y-axis as the proportion. Lines are plotted for all cases, serotype III cases, and controls.
Round 2
Reviewer 2 Report
Revised manuscript has been improved well.
Reviewer 3 Report
The authors have corrected the manuscript that is now suitable for publication.